# DDM1 Maintains Heterochromatin by Regulating Histone Variants

**DOI:** 10.3390/ijms26104845

**Published:** 2025-05-19

**Authors:** Yuanyi Sun, Qijun Xie, Huaixue Chu, Bin Lv, Linan Xie, Qingzhu Zhang

**Affiliations:** 1College of Life Science, Northeast Forestry University, Harbin 150040, China; yuanyi_sun@nefu.edu.cn (Y.S.); qijun_xie@nefu.edu.cn (Q.X.); 2022122919@nefu.edu.cn (H.C.); 2022112931@nefu.edu.cn (B.L.); 2Key Laboratory of Sustainable Forest Ecosystem Management-Ministry of Education, School of Ecology, Northeast Forestry University, Harbin 150040, China; 3State Key Laboratory of Tree Genetics and Breeding, Northeast Forestry University, Harbin 150040, China; 4The Center for Basic Forestry Research, College of Forestry, Northeast Forestry University, Harbin 150040, China

**Keywords:** DDM1, histone variants, histone variant exchange, chromatin remodeling factors, heterochromatin maintenance, TE silencing

## Abstract

Chromatin remodeling factors efficiently and precisely establish, maintain, regulate, and distinguish between chromatin states in eukaryotes. DECREASE in DNA METHYLATION 1 (DDM1) is an important heterochromatin remodeling factor in plants that is responsible for maintaining heterochromatin DNA methylation and suppressing most transposable elements. Previous studies have predominantly focused on the effects of DDM1 on chromatin, with only a few focusing on its remodeling mechanisms. However, recent studies have greatly advanced understanding of the remodeling functions of DDM1 and, in particular, have clarified the mechanisms involved. In this review, we discuss the newly identified remodeling functions and mechanisms of DDM1. As DDM1 is closely involved in histone variant exchange, we first introduce the main histone variants associated with chromatin states in plants. Next, we focus on how DDM1 promotes the deposition of specific histone variants and describe its other remodeling functions. We propose that the core function of DDM1 is the regulation of histone variant distribution. DDM1 maintains heterochromatin by regulating the deposition of H2A and H3 variants, particularly by facilitating the exchange of specific histone variants.

## 1. Introduction

Eukaryotic genomic DNA wraps around histone octamers to form nucleosomes, which serve as the fundamental units that constitute chromatin. Different DNA regions are subject to distinct types of regulation and are located in chromatin regions with different epigenetic states. Chromatin states are defined by epigenetic marks, such as DNA methylation, histone modifications, and histone variants. Distinct histone modifications are associated with different chromatin states [1,2]. An emerging view is that histone variants play a significant role in determining chromatin states, at least as important as, if not more important than, histone modifications [3,4]. Chromatin states are regulated by many intricate and complex remodeling processes, with chromatin remodeling factors playing crucial roles in these processes. Chromatin remodeling factors are a set of proteins that utilize energy from ATP hydrolysis to translocate DNA and exchange or remove specific histone variants, exerting a strong influence on chromatin marks and the dynamic state of chromatin [5,6,7,8]. Different chromatin remodeling factors have different effects on the regulation of chromatin states [7,8], among which DDM1 is an important heterochromatin remodeling factor.

DDM1 is responsible for the maintenance of heterochromatin DNA methylation [9,10,11] and plays a prominent role in silencing most transposable elements (TEs) in plants [11,12]. It functions at pericentromeric heterochromatin regions and transposable elements scattered along chromosome arms [13]. Previously, DDM1 was found to have ATPase activity that can be stimulated by naked and nucleosomal DNA, and recombinant DDM1 is capable of remodeling nucleosomes in vitro [14,15]. DDM1 promotes the deposition of H2A.W in heterochromatin [12] and facilitates the access of DNA methyltransferase to H1-containing heterochromatin [11,16]. However, the mechanisms by which DDM1 remodels nucleosomes, regulates nucleosome properties, and exchanges histone variants were all unclear. Recent studies have provided deep insights into the remodeling functions and mechanisms of DDM1. The exact mechanism via which DDM1 remodels and translocates DNA has been revealed at high resolution [13,17,18,19]. The function of DDM1 in counteracting the low accessibility of DNA in nucleosomes containing H2A.W has also been elucidated [17]. It has also been revealed that DDM1 not only acts on H2A.W but also functions to evict H2A.Z and H3.3 from heterochromatin and deposit H3.1 [4,13,20]. The mechanisms by which DDM1 facilitates histone variant exchange have also been examined to some extent [4,13,17,18,19,20].

In this article, we discuss the remodeling functions and mechanisms of DDM1. Given the close relationship between DDM1 and histone variants, we first summarize the main H2A and H3 histone variants in plants that are associated with different chromatin states. Next, we provide a comprehensive review of new advances in DDM1 research, focusing on the functions of DDM1 in the promotion of the deposition of specific histone variants. The roles of DDM1 in translocating DNA and relaxing nucleosomal DNA ends are also reviewed. Considering all the chromatin remodeling functions of DDM1, we suggest that the core function of DDM1 is the regulation of histone variant distribution.

## 2. The Main Histone Variants Associated with Chromatin States in Plants

Core histones H2A and H3 have multiple variants, with the H2A histone family being the most complex and diverse [21]. In plants, four major H2A types have been identified: H2A, H2A.X, H2A.Z, and H2A.W, and three main H3 histone types: canonical H3.1/H3.2, H3.3, and CenH3 [21,22]. Within a nucleosome, the two H2A histones are usually of the same type and form homotypic nucleosomes, whereas the two H3 histones often coexist as two different types [4]. Histone variants regulate the interaction between the core histone octamers and DNA, conferring specific structural and functional properties to the nucleosomes in which they are incorporated [3,22,23]. Additionally, histone variants can serve as dynamic molecular landmarks for the regulation of histone modifications, cooperatively distinguishing different chromatin landscapes [4]. H2A variants exhibit strong associations with histone modifications, which play a prominent role in determining chromatin states [4], while H3 variants have weaker associations with specific histone modifications [4,24,25].

### 2.1. Euchromatin Histone Variants

The H2A variant H2A.Z and the H3 variant H3.3 are euchromatin histone variants in plants, and neither are replicative histones. H2A.Z is typically deposited into gene bodies and downstream of the transcription start site (TSS) of transcriptionally active genes [26]. In vitro, H2A.Z nucleosome arrays are arranged regularly but not tightly [27]. The impact of H2A.Z on transcription, whether positive or negative, is inconsistent and is regulated by acetylation and ubiquitination, which are associated with developmental stages or environmental signals [26,28,29,30], while the role of H2A.Z in the inhibition of DNA methylation is consistent and clear [20,31]. H3.3 is deposited into transcriptionally active genes to replace H3.1, mainly after cells exit the cell cycle, promoting DNA methylation at gene bodies by repelling linker histone H1 recruitment [4,11,32,33,34]. However, H3.3 does not promote DNA methylation everywhere; rather, it disrupts the maintenance of heterochromatin DNA methylation after it is aberrantly deposited into heterochromatin regions in ddm1 mutants of Arabidopsis thaliana [13]. In addition, H3.3 in euchromatin promotes active histone modifications at some loci, such as histone H3 lysine 4 trimethylation (H3K4me3) and histone H3 lysine 36 trimethylation (H3K36me3) at FLOWERING LOCUS C (FLC) and its homologs [35], and H3K4me3 at lncRNA transcribing regions [36]. Moreover, methylation and acetylation of histone H3 lysine 36 (H3K36) and histone H3 lysine 37 (H3K37) are preferentially associated with H3.3 rather than H3.1 [4]. It remains uncertain whether the association between H3.3 and other epigenetic marks is locus-specific or prevalent genome-wide.

### 2.2. Heterochromatin Histone Variants

The H2A variant H2A.W and the canonical histone H3.1 are enriched in plant heterochromatin. H3.1, a replicative histone, is incorporated into nucleosomes at replication forks [33,37], but only H3.1 deposited in heterochromatin regions remains after cell division [38,39]. After being deposited, H3.1 promotes the recruitment of the histone reader TONSOKU (TSK) to nascent chromatin before chromatin maturation, regulating DNA repair and facilitating the maintenance of several heterochromatin marks, including histone H3 lysine 9 dimethylation (H3K9me2), DNA methylation, H2A.W, and histone H3 lysine 27 trimethylation (H3K27me3) [26,40,41,42]. During cell division, H3.1 maintains histone H3 lysine 27 monomethylation (H3K27me1) and H3K27me3, providing a positive feedback loop that sustains constitutive heterochromatin and polycomb repressive states [4]. The typical heterochromatin mark H3K27me1 is located primarily on H3.1 and plays a crucial role in silencing TEs [43]. Moreover, H3K27me1 inhibits acetylation at H3K27 and H3K36 sites, which contributes to stability within constitutive heterochromatin regions [4,43]. H2A.W is incorporated into chromatin after DNA replication and is only directly deposited into heterochromatin [21,44]. H2A.W plays a more prominent role in maintaining heterochromatin states than H3.1 [4]. H2A.W is often co-deposited with typical heterochromatin marks H3K9me1/2 and H3K27me1 [4,21,44]. Within the nucleosome, two H2A.W molecules interact with each other and with the αN and α2 regions of H3, which stabilizes the docking domains of the two H2A.W molecules as well as the interaction between the αN region of H3 and DNA [17,19]. In addition, the unique KSPKK motif in the C-terminal tail of H2A.W protects the DNA at the entry/exit site of the nucleosome [17,19,44]. Together, these factors cause 145 bps of DNA to tightly wrap around the histone octamer in the H2A.W nucleosome [17]. However, H2A.W inhibits the excessive deposition of linker histone H1 in heterochromatin, which balances the heterochromatin accessibility required for DNA methylation [26,45].

## 3. DDM1 Is Important for the Deposition of Heterochromatin Histone Variants

### 3.1. DDM1 Excludes H2A.Z and Deposits H2A.W in Heterochromatin

Previous studies have generally focused on the effects of DDM1 on H2A.W only [12]. However, new studies have highlighted the role of DDM1 in evicting H2A.Z from heterochromatin [4,20]. By adding the consideration of H2A.Z, we can gain a more comprehensive understanding of the role of DDM1 in regulating the distribution of H2A variants. In the pericentromeric regions of wild-type Col-0 *Arabidopsis*, DDM1 and H2A.W are co-enriched with low levels of H2A.Z [20], whereas in those of *ddm1* mutants, H2A.Z levels are abnormally high, and H2A.W levels are significantly reduced. This is associated with the accumulation of abundant co-transcriptional R-loops and loss of DNA methylation [4,20]. Additionally, in those of *ddm1 h2a.z* double mutants, DNA methylation levels are partially restored, and co-transcriptional R-loop levels are reduced [20]. In *h2a.w* mutants, H2A and H2A.X replace H2A.W at its original location in pericentromeric regions and chromosome arms, with almost no change in H2A.Z levels, increased binding of linker histone H1, and only small changes in DNA methylation levels [45]. The ectopic deposition of H2A.Z results in loss of DNA methylation and accumulation of co-transcriptional R-loops, which poses a greater threat to heterochromatin homeostasis than the loss of H2A.W. DDM1 maintains heterochromatin marks and transcriptional suppression by excluding H2A.Z and depositing H2A.W.

Notably, the ectopic deposition of H2A.Z is not directly associated with transcription in heterochromatin [4,20], as in most regions where H2A.Z is ectopically deposited, its level may be sufficient for transcription initiation but not for sustained transcription elongation. A recent study revealed that the expression levels of a lot of TEs are not affected in *ddm1* homozygous mutant plants segregated from heterozygous *ddm1* mutants. Only one-third of previously non-expressed TEs are expressed in these mutants, and some heterochromatin marks, such as H3K27me3, are still present [4]. It was also found that as H2A.Z further transitions to other H2A variants, more heterochromatin marks are lost, more euchromatin marks, such as H3K36me3, are gradually acquired, and the previous heterochromatin regions ultimately open up fully for transcription [4]. Interestingly, a previous study using the *ddm1-2* mutant line described in 1999 suggested that the majority of TEs in *ddm1* mutants are reactivated for expression [11]. This discrepancy may be because self-fertilized *ddm1* homozygous mutants were used in the previous study. Based on a model for the epigenetic inheritance of unmethylated transposable elements [13], we suppose that as *ddm1* mutants undergo multiple generations of self-crossing, many heterochromatin regions gradually become transcriptionally active and accumulate through epigenetic inheritance.

Recent studies have indicated that DDM1 may directly mediate histone variant exchange to regulate the distribution of H2A variants. In vitro, two previously reported H2A.W-binding domains of DDM1 were recently reported to also bind H2A.Z, but not H2A or H2A.X [4,12,20]. DDM1 has a weaker binding affinity for H2A.Z than H2A.W, and H2A.W can disrupt the interaction between DDM1 and H2A.Z [20]. It is possible that DDM1, owing to its differential binding affinities, first binds to H2A.Z nucleosomes before executing histone exchange after the arrival of H2A.W. In addition, DDM1 allows H2A nucleosomes to adopt an unwrapped conformation [19], which might be the first step in histone variant exchange. This suggests that DDM1 may also directly participate in the exchange of H2A for H2A.W.

### 3.2. DDM1 Promotes the Deposition of H3.1 in Heterochromatin

In *Arabidopsis ddm1* mutants, the levels of H3.1 in chromatin are significantly lower than those in wild-type plants, and H3.3 is ectopically deposited in large quantities in heterochromatin regions [13], suggesting that DDM1 promotes the deposition of H3.1 in heterochromatin. H3.1 is a replicative histone that is incorporated into nucleosomes at replication forks. The replacement of H3.1 with H3.3 takes place after DNA replication and relies on the histone chaperones HIRA (histone cell cycle regulator), which functions broadly during interphase of cell division, and the *Arabidopsis thaliana* ATRX (Alpha-Thalassemia X-Linked Intellectual Disability Syndrome) protein ortholog, which functions only for a short period before mitosis [38,39]. It is possible that as long as the level of DDM1 is sufficient to impede the replacement of H3.1 by H3.3 after DNA replication, it can ensure the stability of heterochromatin regions without being affected by H3.3. Therefore, the exchange of H3.3 back to H3.1 is considered unnecessary for the prevention of ectopic deposition of H3.3 by DDM1. However, DDM1 may be involved in H3 variant exchange, since it possesses an ATP-dependent function that enables nucleosome unwrapping [13]. DDM1 displays weak unwrapping activity toward H3.1 nucleosomes but stronger activity toward H3.3 nucleosomes [13]. After incubating with DDM1, H3.3/H2A nucleosomes and H3.3/H2A.W nucleosomes undergo large-scale unwrapping in vitro [13]. The mechanism via which DDM1 specifically promotes the deposition of H3.1 in heterochromatin is still unclear.

Interestingly, in *ddm1 hira* double mutants of *Arabidopsis*, the levels of DNA methylation are restored to some extent compared with the levels in *ddm1* mutants, with CHH context methylation reaching nearly wild-type Col-0 levels [13]. This finding indicates that H3.3 prevents DNA methylation after its ectopic deposition in heterochromatin [13]. H3.3 promotes DNA methylation at gene bodies in euchromatin by repelling linker histone H1 recruitment, and it is independent of histone modification [34]. However, H3.3 may prevent DNA methylation via a pathway related to histone modifications in heterochromatin, given that H3K36me3 shows a certain preference for H3.3 [4].

## 4. DDM1 Has Other Remodeling Functions

DDM1 interacts with DNA, inducing conformational changes in nucleosomal DNA and driving DNA translocation [13,17,18,19] via a shared mechanism similar to that of yeast Snf2 [6,46,47]. The activities of DDM1 in DDM1-nucleosome complexes containing different H2A histone variants are similar [17]. DDM1 structural analysis and several working models have provided deep insights into the mechanism of DDM1 (Figure 1a). Nucleotide-free DDM1 possesses an open conformation [18]. After it binds to the nucleosome at the superhelical location 2 (SHL2), it unwinds and pulls the upstream DNA, causing an opening between two DNA gyres and generating a nucleosomal DNA bulge [13,17,18,19]. The addition of ATP changes the conformation of DDM1 from an “open” to a “closed” configuration, which results in the release of nucleosomal DNA bulges and a more relaxed double-helix state [18]. Simultaneously, DDM1 Lobe2 anchors the DNA and pushes it toward DDM1 Lobe1, which couples to the rotational movement of Lobe1 and squeezes the DNA toward its downstream exit side [18]. Upon ATP hydrolysis, DDM1 reopens, pulling the incoming DNA from the entry side and generating a nucleosomal DNA bulge once again [18]. After ADP is released, DDM1 undergoes no further conformational change and directly enters the next remodeling cycle [18,19]. DDM1 remodels and translocates DNA by cyclically hydrolyzing ATP and undergoing conformational changes.

DDM1 also interacts with the C-terminal tail of H2A.W, without the need for ATP hydrolysis [17,19]. This reorganizes the interaction between DNA and the histone octamer within a nucleosome, increasing the flexibility of the entry/exit nucleosomal DNA ends (Figure 1b) [17]. In the H2A.W nucleosome, 145 bps of DNA are tightly wrapped around the histone octamer, whereas after DDM1 binding, only 111 bps of DNA remain wrapped around the histone octamer, similar to what is observed in the H2A nucleosome [17]. This change counteracts the low DNA accessibility caused by H2A.W [17] and probably reduces resistance to DNA translocation in the H2A.W nucleosome. This may explain why DDM1 translocates DNA equally well in DDM1-nucleosome complexes containing different H2A variants. On the other hand, this effect may directly increase the accessibility of DNA in constitutive heterochromatin, which would allow the entry of DNA methyltransferase and histone-modifying complexes [17].

## 5. The Core Function of DDM1 Is the Regulation of Histone Variant Distribution

The function of DDM1 is closely related to histone variants. In heterochromatin, DDM1 is essential for the deposition of H2A.W and H3.1 and the eviction of H2A.Z and H3.3. It is unreasonable to consider the function of DDM1 outside the context of histone variants or to examine the roles of heterochromatin histone variants without taking DDM1 into account. Thus, to provide a more comprehensive view, we discuss below the role of DDM1 in regulating H2A and H3 variants together. In the absence of DDM1, ectopic deposition of H2A.Z and H3.3 is observed, which prevents DNA methylation and leads to a drastic reduction in H3K9me1/2 and a moderate increase in H3K27me3 [4,11,12]. Together, these changes loosen the nucleosome arrangement and promote the initiation of transcription in heterochromatin. Furthermore, H2A.Z is unable to impede the deposition of euchromatin marks [4] and is easily exchanged for other histone variants [4,26,27]. This leads to imbalances in nucleosome properties and the accumulation of euchromatin marks, particularly H3K36me3, which makes heterochromatin prone to be involved in transcription (Figure 2a) [4,35]. When DDM1 is present, H2A.W and H3.1 are accurately deposited into heterochromatin, ensuring the stability of heterochromatin properties [4,13,20,45]. Within nucleosomes, H2A.W and H3.1 interact strongly with DNA, constraining the entry/exit of nucleosomal DNA ends [17,19]. In heterochromatin regions, H2A.W and H3.1 contribute to the compaction of chromatin structure, preventing euchromatin marks but not heterochromatin marks, causing high levels of H3K9me2 and H3K27me1 (Figure 2b) [4,12,13,44]. Moreover, given that the ectopic deposition of H2A.Z and H3.3 in heterochromatin significantly prevents DNA methylation, and DDM1 can efficiently exchange histone variants, stable DNA methylation in heterochromatin regions is likely to occur only after DDM1 has regulated the exchange of H2A and H3 variants.

DDM1 may directly participate in the process of histone variant exchange because it has certain functions, such as interacting with H2A.W and H2A.Z [4,12], allowing H2A nucleosomes to adopt an unwrapped conformation [19], unwrapping H3.3 nucleosomes [13], and increasing the flexibility of nucleosomal DNA ends in H2A.W nucleosomes [17]. These functions lead to a histone variant exchange model of DDM1 (Figure 3). Additionally, the ortholog of DDM1 in mammals, the chromatin remodeling factor lymphoid-specific helicase (LSH), directly catalyzes the exchange of H2A variants through ATP hydrolysis [48]. LSH mostly exists as a monomer, and similar to LSH, DDM1 is not found in any chromatin remodeling complexes, although DDM1 may synergistically operate with DNA methyltransferases [13,49]. It is probable that DDM1, similar to LSH, acts independently of other factors and directly catalyzes the exchange of histone variants.

In addition to directly participating in the exchange of H2A and H3 histone variants, other functions of DDM1 may promote the deposition of specific histone variants. DDM1 is known to remodel and translocate DNA in heterochromatin [13,17,18,19], but it remains unclear whether the resulting sliding of the nucleosome loosens or tightens chromatin. In a sense, the exact effect of DDM1 on the degree of chromatin compaction may be complex and relative. If the range and length of chromatin are limited, when one part of the chromatin becomes tighter, the other parts will inevitably become looser. DDM1 may first recognize heterochromatin regions and slide nucleosomes to increase chromatin accessibility before recruiting other proteins for histone variant exchange. Alternatively, DDM1 may act as a chaperone protein, first participating in the formation of a remodeling complex to carry a histone variant or carrying a histone variant on its own, and then leading it to access DNA. However, another possible role of DDM1 could be to prevent the replacement of H3.1 deposited at replication forks by compacting the heterochromatin nucleosome array through translocating DNA.

## 6. Conclusions and Future Perspectives

The currently known mechanisms of DDM1 suggest that DDM1 maintains heterochromatin by regulating histone variant distribution, particularly by facilitating the exchange of specific histone variants. DDM1 acts on both H2A and H3 variants, which distinguishes it from usual chromatin remodeling factors or histone chaperones. Moreover, DDM1 not only regulates histone variant distribution but also affects interactions between histone octamers and DNA, conferring specific properties to heterochromatin nucleosomes. However, some aspects of the remodeling of histone variants by DDM1 remain unclear, and others need further confirmation. For instance, it is unclear whether DDM1 exercises catalytic activity on histone variant exchange, although it is considered to participate directly in this process. This could initially be investigated using the histone transfer assay that was previously used to discover the remodeling activity of LSH in vitro [48] and then by constructing specific systems to confirm the results of this assay in vivo. Additionally, the interaction between DDM1 and H3 variants requires confirmation. It is unclear whether DDM1 interacts with H3.1 and H3.3 histone variants, since the results of mass spectrometry and cryo-EM analyses have been inconclusive [13,17,20]. If DDM1 does not interact with H3.1 and H3.3, then it cannot directly mediate the exchange of H3 histone variants. It is generally accepted that DDM1 is directly responsible for the exchange of H2A variants and that it cooperates with other proteins to facilitate the exchange of H3 variants. The mechanism responsible for the specific remodeling function of H3.3 nucleosomes of DDM1 remains unclear as well. It may involve yet unknown domains that determine the direction of chromatin remodeling activity, as it has been shown that the N-terminal (1-132) truncated form of DDM1 also strongly remodels H3.1/H2A nucleosomes [13]. Answers to these outstanding issues might be forthcoming once the interaction between DDM1 and H3 variants has been clearly defined in the future.

Chromatin remodeling factors and chromatin marks together constitute a complex network for the regulation and maintenance of chromatin states. A chromatin remodeling factor can remodel multiple chromatin marks at a time, regulating the formation or maintenance of chromatin states dynamically and efficiently. Previously, chromatin remodeling factors that have been extensively studied in plants are mainly those involved in regulating gene expression, plant organ development, and DNA damage repair, such as inositol requiring 80 (INO80), SWI2/SNF2-related 1 (SWR1), chromodomain helicase DNA-binding (CHD1), and E1A binding protein p400 (EP400) [50,51,52,53,54,55]. Recent research into DDM1 function has increased our understanding of heterochromatin remodeling factors, but the mechanisms involved in the regulation of heterochromatin dynamics are still not fully understood. It is expected that as the functions and remodeling mechanisms of more chromatin remodeling factors are revealed in plants, the network governing the regulation of chromatin state maintenance will become clearer. These will provide a broader theoretical foundation for studying chromatin dynamics in important plant varieties as well as references for epigenetic studies in humans and animals.

## Figures and Tables

**Figure 1 ijms-26-04845-f001:**
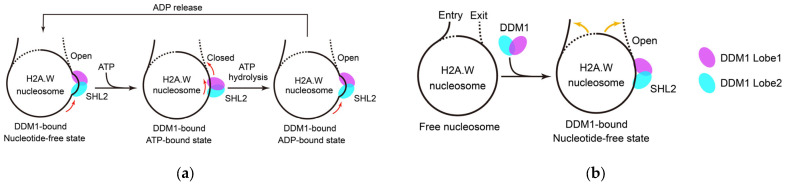
The remodeling functions of DDM1 to nucleosomal DNA in heterochromatin. (**a**) DDM1 remodels and translocates DNA by cyclically hydrolyzing ATP and undergoing conformational changes; (**b**) DDM1 increases the flexibility of the entry/exit nucleosomal DNA ends, which counters the low DNA accessibility caused by H2A.W. Red arrows indicate the possible directions of DNA translocation; yellow arrows represent the movements of nucleosomal DNA ends.

**Figure 2 ijms-26-04845-f002:**
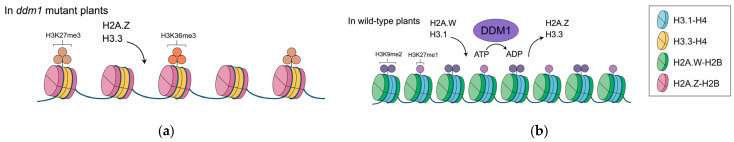
The remodeling function of DDM1 in depositing specific H2A and H3 variants in *Arabidopsis thaliana*. (**a**) In *ddm1* mutant plants, H2A.Z and H3.3 are ectopically enriched in heterochromatin, with a moderate increase in H3K27me3, and there may be incorporation of H3K36me3; (**b**) In wild-type plants, DDM1 evicts H2A.Z and H3.3 and promotes the deposition of H2A.W and H3.1 in heterochromatin, facilitating the enrichment of H3K9me2 and H3K27me1.

**Figure 3 ijms-26-04845-f003:**
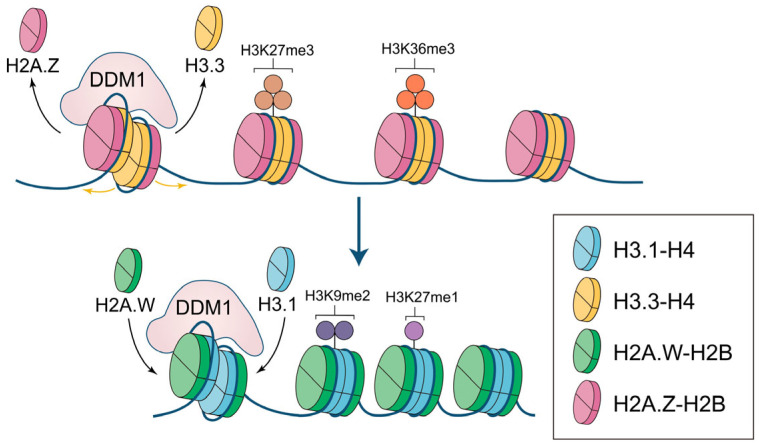
The histone variant exchange model of DDM1. DDM1 evicts H2A.Z and unwraps H3.3 nucleosomes and promotes the deposition of H2A.W and H3.1. The function of DDM1 in counteracting the low accessibility of DNA in nucleosomes containing H2A.W may promote the maintenance of H3K9me2 and H3K27me1 subsequently. DDM1 ensures the stability of repressive marks in heterochromatin.

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
