# Peer review of "DDM1 Maintains Heterochromatin by Regulating Histone Variants"

_ijms, 2025, doi:10.3390/ijms26104845_

Round 1
Reviewer 1 Report
Comments and Suggestions for Authors
This review highlights recent insights into the chromatin remodeling functions of DDM1, a key plant factor maintaining heterochromatin DNA methylation and transposon silencing. It emphasizes DDM1’s role in histone variant exchange, particularly H2A and H3 variants, as a central mechanism in chromatin state regulation. The authors propose that DDM1's core function is controlling histone variant distribution to maintain chromatin structure and function.
Major adjustment needed.
Line 47-48; The authors mention that DDM1's remodeling mechanisms were unclear—what specific knowledge gaps still remain despite recent advances?
Line 54; Could the authors clarify how DDM1 overcomes the low accessibility of H2A.W-containing nucleosomes?
Line 57-58; On what basis do the authors suggest that deposition of specific histone variants is the primary function of DDM1 over others like DNA translocation?
Line 65-67; What experimental strategies could be used to test the hypothesis that histone variant regulation is DDM1’s core function?
Line 71-72; You mention "replicative H2A" but do not define or elaborate on its function. Could you clarify its specific role and how it differs from other H2A variants?
Line 74; The claim that H3 histones often coexist as different types in the same nucleosome is intriguing. Can you provide experimental evidence or references supporting this heterotypic pairing in vivo?
Line 80-81; It is stated that H3 variants have weaker associations with histone modifications. Is this universally true across plant species, or are there exceptions in certain tissues or developmental stages?
Line 86-88; The dual role of H2A.Z in transcription is said to be "inconsistent." Can this be contextualized with developmental stage or environmental signals to improve clarity?
Line 97-98; The suppression of H3K27me3 by H3.3 at FLC is mentioned. Could the authors elaborate on whether this regulatory role is locus-specific or more widespread?
Line 103-104; H3.1 is said to promote recruitment of TSK before chromatin maturation. What experimental technique was used to determine this specific temporal window of recruitment?
Line 114-115; The authors claim H2A.W has a more prominent role in heterochromatin maintenance than H3.1. What comparative quantitative evidence supports this conclusion?
Line 121-122; The authors mention that 145 bps of DNA wrap tightly around H2A.W nucleosomes. Does this differ from canonical nucleosomes (147 bps)? If so, how does this affect nucleosome stability?
Line 148-160; Could the authors provide experimental evidence or citation supporting the claim that multiple generations of selfing in ddm1 mutants leads to cumulative heterochromatin opening? Is there a known mechanism of epigenetic drift over generations in ddm1 mutants that supports this hypothesis? How do the authors reconcile this with the observation that only one-third of previously silenced TEs become expressed in newer studies?
Line 171-200; The section lacks mechanistic detail explaining how DDM1 preferentially promotes H3.1 deposition while avoiding H3.3 accumulation in heterochromatin. Do the authors have any chromatin immunoprecipitation (ChIP) data showing co-localization of DDM1 with H3.1 but not H3.3? How does the unwrapping preference of DDM1 toward H3.3-containing nucleosomes (line 186) mechanistically explain the reduced H3.3 levels in wild-type? Could this preferential unwrapping facilitate H3.3 eviction?
Line 194-199; The manuscript notes a contradiction: H3.3 inhibits methylation in heterochromatin but promotes it in euchromatin. Could the authors explore whether histone PTMs (e.g., H3K36me3) that accompany H3.3 deposition differ between chromatin contexts and modulate DNA methyltransferase access?
A clear figure needed - Add a schematic summarizing the histone variant exchange model by DDM1, including H2A.Z eviction, H2A.W and H3.1 deposition, and their downstream epigenetic effects.
Line 218-227; The manuscript suggests ATP-independent remodeling via DDM1-H2A.W interaction, but doesn’t connect this clearly to biological function. How does the ATP-independent partial unwrapping of DNA around H2A.W by DDM1 relate to DNA methylation maintenance or TE silencing in vivo?
Revise the reference; add the references from the current decade.
Reviewer 2 Report
Comments and Suggestions for Authors
I read the review written by Sun et al. Role of DDM1 and histone variants, and their relationship are nicely reviewed. I have several comments:
- Authors described "H3.1, a replicative histone, is incorporated into nucleosomes at replication forks, but only H3.1 deposited in heterochromatin regions remains after cell division [28,29]" (line 102). However, both ref. 28 & 29 did not mention that H3.1 is a replicative histone. In animals, H3.1 was confirmed to be a replicative histone, but I do not know whether H3.1 is a replicative histone in plants. If so, authors should add appropriate citation(s). Several sentences mentioned that H3.1 is a replicative histone (in plants) (e.g. line 175).
- As authors know I think, the function of H3K27me3 is quite different from that of H3K27me1, H3K9me2, and DNA methylation. H3K27me3 is involved in facultative heterochromatin and tissue- or stage-specific gene expression. H3K27me1, H3K9me2, and DNA methylation are involved in constitutive heterochromatin and stable repression of TEs and a part of genes. For general readers who is beginner for this field, authors should describe this difference early in the review. This helps readers understand the different behavior of H3K27me3 and H3K9me2/H3K27me1/DNA methylation in ddm1 mutant depicted in Fig. 2.
- Authors stated "the expression levels of most TEs are not affected in ddm1 homozygous mutant plants segregated from heterozygous ddm1 mutants. Only one-third of previously non-expressed TEs are expressed in these mutants" (line 148). These two sentences seem to be controversial. "one-third of previously non-expressed TEs are expressed (in ddm1)" means the expression levels of a part of TEs are affected in ddm1. "most" seems to be too strong to describe the phenotype.
- "Only when DDM1 has exchanged histone variants can the properties of heterochromatin regions be stabilized" (line 239) makes no sense. Please revise it.
- In Author Contributions section, Authors stated "funding acquisition, L.X. and Q.Z." (line 332). As MDPI authorship criteria and ICMJE authorship criteria, funding acquisition is not qualified as the authorship. Please delete it.
Round 2
Reviewer 1 Report
Comments and Suggestions for Authors
The author has revised the MS as per the suggestions.